# Mycobacteriophage Alexphander Gene *94* Encodes an Essential dsDNA-Binding Protein during Lytic Infection

**DOI:** 10.3390/ijms25137466

**Published:** 2024-07-07

**Authors:** Emmanuel Chong Qui, Feben Habtehyimer, Alana Germroth, Jason Grant, Lea Kosanovic, Ivana Singh, Stephen P. Hancock

**Affiliations:** Department of Chemistry, Towson University, Towson, MD 21252, USA; emmanuelecq2@gmail.com (E.C.Q.); habtehyimerfeben@gmail.com (F.H.); alanagermroth@gmail.com (A.G.); jasongrant0512@gmail.com (J.G.); lk882@georgetown.edu (L.K.); ivanasingh96@gmail.com (I.S.)

**Keywords:** mycobacteriophage, DNA-binding protein, helix–turn–helix motif, superinfection

## Abstract

Mycobacteriophages are viruses that specifically infect bacterial species within the genera *Mycobacterium* and *Mycolicibacterium*. Over 2400 mycobacteriophages have been isolated on the host *Mycolicibacterium smegmatis* and sequenced. This wealth of genomic data indicates that mycobacteriophage genomes are diverse, mosaic, and contain numerous (35–60%) genes for which there is no predicted function based on sequence similarity to characterized orthologs, many of which are essential to lytic growth. To fully understand the molecular aspects of mycobacteriophage–host interactions, it is paramount to investigate the function of these genes and gene products. Here we show that the temperate mycobacteriophage, Alexphander, makes stable lysogens with a frequency of 2.8%. Alexphander gene *94* is essential for lytic infection and encodes a protein predicted to contain a C-terminal MerR family helix–turn–helix DNA-binding motif (HTH) and an N-terminal DinB/YfiT motif, a putative metal-binding motif found in stress-inducible gene products. Full-length and C-terminal gp94 constructs form high-order nucleoprotein complexes on 100–500 base pair double-stranded DNA fragments and full-length phage genomic DNA with little sequence discrimination for the DNA fragments tested. Maximum gene *94* mRNA levels are observed late in the lytic growth cycle, and gene 94 is transcribed in a message with neighboring genes *92* through *96*. We hypothesize that gp94 is an essential DNA-binding protein for Alexphander during lytic growth. We proposed that gp94 forms multiprotein complexes on DNA through cooperative interactions involving its HTH DNA-binding motif at sites throughout the phage chromosome, facilitating essential DNA transactions required for lytic propagation.

## 1. Introduction

Bacteriophages (or phages) are viruses that infect bacteria. Phages are immensely diverse, display a wide array of host-specificity profiles, and employ numerous molecular mechanisms through which they interact with their bacterial targets [1]. Phages comprise the most numerous biological entities on planet Earth [2], and it is hypothesized that, collectively, they can infect every bacterial species, with each phage targeting a specific group of related bacterial organisms. Consequently, phages are important drivers of bacterial community evolution and microbiome composition [3]. The study of phage–host interactions has been key to elucidating the central dogma of molecular biology and has generated a myriad of biotechnology and therapeutic tools that impact a wealth of industrial and biomedical processes, from food safety to treatment and detection of antibiotic-resistant bacterial infections [1,4,5,6,7]. Moreover, it is widely accepted that continued investigation into the molecular mechanisms phages employ to optimally interact with host cells under various environmental conditions will uncover novel phage-based applications for use in industry and biomedicine.

Mycobacteriophages infect members of the genera *Mycobacterium* and *Mycolicibacterium,* which includes pathogenic and nonpathogenic organisms, such as *Mycobacterium tuberculosis* and *Mycolicibacterium smegmatis*. Over 2400 mycobacteriophages have been isolated and sequenced, many in the past two decades, due in large part to the “Phage Hunters Integrating Research and Education” (PHIRE) and “Science Education Alliance-Phage Hunters Advancing Genomics and Evolutionary Science” (SEA-PHAGES)—two discovery-based high school/undergraduate research programs that began at the University of Pittsburgh [8] and have now been implemented at over 200 institutions internationally [9,10]. The wealth of genomic data generated by these efforts reveals remarkable diversity and mosaicism within mycobacteriophage genomes, likely due to various recombinatorial events, including horizontal gene transfer, illegitimate recombination, and site-specific integration, between and within the bacterial host [1,11]. Sequenced mycobacteriophages are grouped into thirty-one clusters and seventy-three subclusters, with seven phages having insufficient homologous gene content to be grouped in any current cluster (singletons). Over half of the mycobacteriophage-encoded genes lack a suitable ortholog and have no known function. This makes functional gene annotation difficult. Additionally, many genes that have been assigned a putative function using homology algorithms remain uncharacterized. Identifying the function of these genes and determining how they control the phage lifecycle and potentially mediate phage–host interactions is, therefore, a significant area of interest in phage biology, and the landscape for novel discovery is vast.

Rigorous and elegant studies have begun to evaluate the roles of these unknown gene products using bioinformatics, biochemical and biophysical analysis, mutational analysis, biological toxicity, and/or omics-based approaches [12,13,14,15,16]. In one study, systematic gene deletion and expression profiling of mycobacteriophage Giles found that 45% of genes are dispensable for lytic growth, and every predicted gene product is expressed either during the lytic or lysogenic life cycles [12]. Additionally, the mycobacteriophage Giles protein–protein interactome has been mapped and has been used to infer functions of unknown proteins in tail assembly and DNA replication [13]. These global analyses are essential for understanding the relationships between phage gene products and phage propagation and are invaluable in helping to assign functions to uncharacterized gene products. However, many questions remain, highlighting the need for studies that directly investigate the biochemical and biological roles of the mycobacteriophage proteome.

Mycobacteriophage Alexphander was isolated, sequenced, and annotated as part of a course-based undergraduate research experience (CURE) at the University of California, Los Angeles, as part of the SEA-PHAGES program (accession # MG962361.1). Alexphander is a member of cluster “F1”, and its genome contains 57,734 base pairs (bp) that encode 104 predicted genes (phagesdb.org [10] accessed on 02-07-2024). Alexphander encodes an integrase and immunity repressor, suggesting that it is a temperate phage whose lifecycle includes both a lysogenic and a lytic phase; however, this has not been demonstrated experimentally until this study.

Approximately 60% of the predicted Alexphander gene products remain unannotated, have no known function, and, to our knowledge, those that have been annotated have not been validated experimentally. Alexphander gene *94* is a representative of gene “phamily” (pham) 165,933, which is encoded by numerous Mycobacteriophages and Gordoniaphages and is broadly annotated to encode a helix–turn–helix-containing DNA-binding protein. Gene *94* orthologs are also expressed in numerous other species within the actinobacteria. Our research group is interested in understanding how DNA-binding proteins of unknown function support lytic phage growth. We are thus compelled to investigate the essentiality and biochemical properties of gene *94* and its gene product. We view gene *94* as interesting because it is present within numerous actinobacteria and their phages, and its domain architecture is unique relative to other proteins containing a similar DNA binding motif. In Alexphander, gene *94* is specifically annotated as containing a MerR-type helix–turn–helix (HTH) DNA-binding motif (phages.org accessed on 02-07-2024), which is characterized as a winged HTH motif (wHTH) that contains a three or four α-helical bundle and an adjacent β-turn–β-wing motif (InterPro). MerR-type HTH-containing proteins can perform various functions and include the bacterial chromosome anchoring protein, RacA [17], and the transcriptional regulator CueR [18]. For these proteins, the HTH motif is in the N-terminal region of the protein, whereas it is C-terminal in gene *94*. Perhaps this indicates that the product of gene *94*, gp94, has a distinct functional role as well. Indeed, we see that Alexphander gp94 displays several features distinct from known MerR-type HTH-containing proteins. Here, we show that gp94 contains a C-terminal HTH DNA-binding motif, behaves as a monomer in solution, and binds double-stranded DNA nonspecifically to assemble large nucleoprotein complexes. These features distinguish gp94 from other MerR-like HTH-containing proteins that have been characterized thus far. In this study, we confirm that Alexphander is a temperate phage, and we use targeted reverse-genetics and biochemical approaches to elucidate aspects of gp94 function related to its role in *M. smegmatis* infection.

## 2. Results

### 2.1. Mycobacteriophage Alexphander Is a Temperate Phage That Forms Stable Lysogens

Mycobacteriophage Alexphander is predicted to be a temperate phage that can transition between lytic and lysogenic replication within *M. smegmatis* strain mc^2^ 155. Here, we find that Alexphander forms clear plaques after 18 h that develop turbid halos after 32 h at 37 °C when spotted on or added to top agar lawns seeded with *M. smegmatis* mc^2^ 155 (Figure 1a). Such halos have been attributed to the diffusion of phage tail fibers that depolymerize peripheral host cell surface membrane structures [19,20]. High titer spots further develop mesas—areas of bacterial regrowth within the circular clearing that are a hallmark feature of lysogenic phages—beginning after 4 to 5 days of incubation. Streaks from these mesas produced stable lysogens that were subjected to three rounds of single-colony purification. Lysogens were verified following each round of purification by testing for phage release and immunity to Alexphander infection (Figure 1b,c). Notably, Alexphander lysogens are also immune to superinfection by the uncharacterized temperate phage LilPharaoh (Cluster K1) and the lytic phage SchoolBus (Cluster O) but sensitive to superinfection by phages Ph8s (Cluster A) and Sheila (Cluster B). The authors note that the low SchoolBus titer in Figure 1c may prohibit the evaluation of subtle differences in infectivity between SchoolBus and LilPharaoh. The authors also note that higher titer SchoolBus stocks are also completely unable to infect the Alexphander lysogen, indicating robust superinfection protection against SchoolBus infection. The breadth of superinfection specificity will require testing a larger library of phages for the ability to infect the Alexphander lysogen. The lytic repression mechanism for LilPharaoh is uncharacterized and, given the low level of amino acid sequence identity between the putative repressors encoded by LilPharaoh and Alexphander (24.6% with an “Emboss WATER” pairwise alignment score of 32.0) and the low level of nucleotide conservation between Alexphander and SchoolBus, it is likely that the Alexphander lysogen mediates heterolytic protection against LilPharaoh and SchoolBus infection through a yet uncharacterized mechanism that is not mediated by the Alexphander immunity repressor.

We measured the efficiency of Alexphander lysogeny by plating naïve *M. smegmatis* mc^2^ 155 onto uninfected 7H10 agar plates or those seeded with Alexphander. We find that the rate of lysogeny is approximately 3%, which is comparable to, albeit somewhat lower than, that of phage Giles (7%), a temperate mycobacteriophage in cluster Q [12]. We note that several other Cluster F1 phages form stable lysogens [15,21]; however, to our knowledge, lysogeny efficiency for these phages has not been explicitly reported. Despite the relatively low rate at which lysogens are formed, Alexphander lysogens are stable, as we observe no difference in the lysogen survival when plated onto phage-seeded vs. unseeded solid media (Figure 1d).

### 2.2. Alexphander Gene 94 Orthologs Are Present within Numerous Actinobacteria and Their Phages

The gene *94* coding sequence is a member of pham 165,933 and overlaps neighboring genes *93* and *95* by four nucleotides (Figure 2a,b and [10]). Pham 165,933 is present in 126 mostly temperate phages (phagesDB.org accessed on 02-07-2024), including 2 different Mycobacteriophage clusters (F and N), 6 Gordoniaphage clusters (CY, CZ, DI, DN, DW, and DY), and 3 singleton phages [22]. In many of these phages, pham 165,933 is more generally annotated to encode helix–turn–helix (HTH) DNA-binding motif-containing proteins. In Alexphander, this pham is annotated to encode a protein containing a MerR-type HTH motif [22]. A PSI-BLAST search shows that gene 94 orthologs are encoded in >113 species within the Actinobacteria and their phages, with an E value < 1 × 10^−33^. Multiple and pairwise sequence alignments of pham 165,933 and its orthologs identify regions of amino acid conservation and show that cluster F variants are 99% identical but only 60 and 30% identical to cluster N mycobacteriophage and Gordonia phage variants, respectively (Figure 2c,d). There are significant regions of identity over the length of the protein sequence, including in the putative DNA-recognition helix, α7, suggesting that these proteins may have similar DNA sequence preference profiles. Interestingly, synteny of the gene *94* locus is conserved within but varies between clusters F and N phages (Figure 2b). For example, in F1 phages (e.g., Alexphander and Phasih), Alexphander gene *94* (Phasih gene 90) is located within a conserved four-gene locus that includes phams 815 (gene *93*/*89*), 2998 (gene *95*/*91*), and 371 (gene *96*/*92*); however, these genes are absent in pham 165,933 containing phages in clusters N and CZ.

### 2.3. The Mycobacteriophage Alexphander Gene 94 Is Essential for Lytic Alexphander Infection

The functions of approximately 60% of predicted Alexphander gene products are unannotated, and, to our knowledge, none of the annotated gene product functions have been characterized experimentally. There are numerous Alexphander genes that are predicted to encode DNA-binding motif-containing proteins. Here, we performed a deletion analysis of Alexphander gene *94* to determine its essentiality for lytic growth. We used the recombineering tool, Bacteriophage Recombineering of Electroporated DNA (BRED) [24], to delete gene *94*. Briefly, a recombineering strain encoding orthologs of λ red recombination enzymes from a plasmid (mc^2^ 155*::*pJV53) is co-electroporated with phage genomic DNA (gDNA) and a 400 bp recombination substrate, which contains 200 bp of DNA up and downstream of the site to be deleted. Electroporated cells are then plated with naïve *M. smegmatis* mc^2^ 155, mc^2^ 155 containing an empty expression vector (mc^2^ 155 (pLAM12)), or mc^2^ 155 containing a complementation plasmid providing exogenous gp94 from an acetamide-inducible promoter (mc^2^ 155(pSH20)). Recombination between the gDNA and substrate results in the elimination of nucleotides 52527 to 53159 (92% of the gene) but leaves the twenty-seven nucleotides at both the 5′ and 3′ ends of the coding sequence, which contain the stop codon of overlapping gene *93* and potential regulatory sequences upstream of gene *95* (Figure 2a). Mixed phage populations containing recombinant and wild-type WT alleles were observed in primary plaques harvested from mc^2^ 155 and mc^2^ 155(pSH20) as measured by flanking PCR assays (Figure 3a–c). Recombinant phages were not observed in plaques harvested from mc^2^ 155 (pLAM12) (Figure 3b,c). Interestingly, recombination frequency and the intensity of the recombinant amplicon increase significantly when plaques are harvested from the complementation strain (9/35 vs. 1/35), suggesting that the presence of exogenous gp94 is supporting mutant phage growth. We were able to isolate pure gene *94* deletion mutants containing only the recombinant allele only when primary mixed plaques were plated onto the complementation strain. No recombinant phage was present in plaques or lysates collected from non-complementing strains, suggesting that gene *94* is essential (Figure 3d). To verify its essentiality for lytic growth, we measured the ability of WT and Δ*94* Alexphander to plaque on *M.* smegmatis mc^2^ 155, mc^2^ 155 (pLAM12), or the complementation strain, mc^2^ 155(pSH20). WT phages can plaque on all three host strains, whereas Alexphander Δ94 can only plaque on the complementation strain (Figure 3e). This provides clear evidence that gene 94 is essential for Alexphander lytic growth.

**Figure 3 ijms-25-07466-f003:**
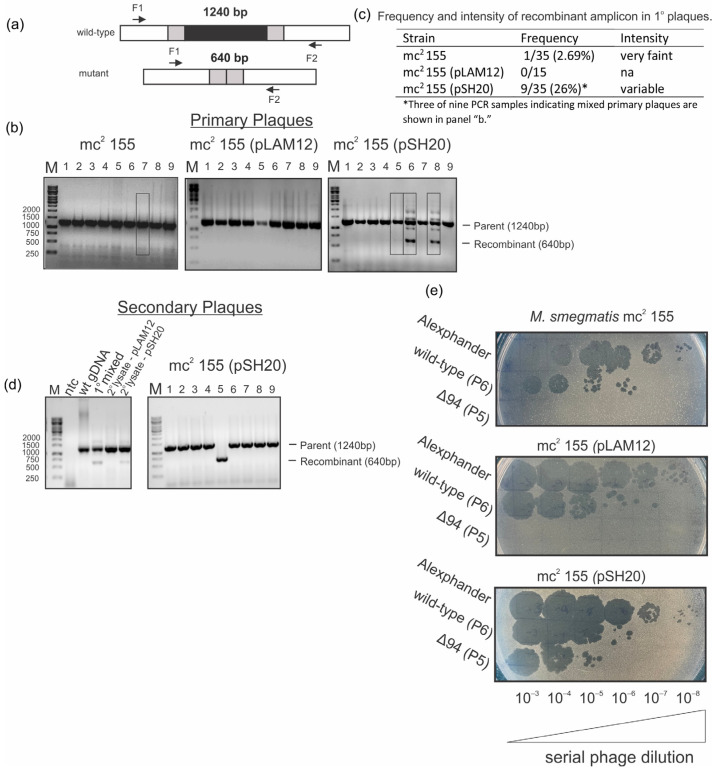
Gene 94 is essential for Alexphander lytic growth. (**a**) Schematic of the flanking PCR strategies for deletion allele screening. Flanking primers F1/F2 recognize genomic sites flanking the gene 94 locus as indicated. PCR amplification of recombinant phages will result in smaller amplicons for the gene 94 deletion (640 bp), as indicated. (**b**) Representative gels indicating the presence of the recombinant deletion strain in mixed primary plaques following co-electroporation of naïve host (far left), host containing control empty plasmid (pLAM 12; middle), or host containing the complementation plasmid (pSH20; far right). The recombineering strain, *M. smegmatis* mc^2^ 155::pJV53, was co-electroporated with 200 ng of Alexphander gDNA and 150 ng of a 400 bp gene 94 deletion substrate and plated as top agar lawns with naïve *M. smegmatis*
^mc2 155^, mc^2^ 155 containing empty plasmid (mc^2^ 155 (pLAM12)), or mc^2^ 155 containing an acetamide inducible complementation plasmid from which exogenous copies of gp94 are produced (mc^2^ 155(pSH20)) (Table 1). The resulting primary plaques were screened by flanking PCR using primers in Table 2. Mixed plaques containing recombinant phages were identified by the presence of the 640 bp fragment. Recombinant fragments were rare and faint when electroporated cells were plated on *naïve* mc^2^ 155 and were absent when plated on the empty plasmid strain. Recombinant phages were more frequent, and amplification was more robust when electroporated cells were plated onto the complementation strain. (**c**) Table summarizing the frequency and intensity of recombinant amplicons. Note that only three of nine PCR samples indicating the presence of mixed plaques are shown in panel “b”. (**d**) Representative agarose gel indicating the presence of the recombinant deletion strain in mixed secondary lysates or in pure secondary plaques as measured by the presence of a 640 bp PCR amplicon. Mixed primary plaques were plated for purification on the empty plasmid strain (mc^2^ 155 (pLAM12)) or the complementation strain (mc^2^ 155(pSH20)). (Left panel) Recombinant phages were observed in lysates collected from the complementation strain but not from the empty plasmid strain, as indicated by the presence of the recombinant fragment. Controls show no amplification in the absence of a template and a robust parental fragment amplified from purified wild-type Alexphander genomic DNA. (Right panel) Individual plaques contain either parental or mutant phage DNA, indicating that we have successfully purified a Δ*94* mutant phage. (**e**) Gene *94* essentiality was tested by measuring plaquing by wild-type Alexphander and 2° plaques containing parental (P6) or Δ*94* (P5) phages. Plaquing was measured on naïve wild-type *M. smegmatis* mc^2^ 155, mc^2^ 155 (pLAM12), and (mc^2^ 155(pSH20). Wild-type Alexphander can plaque on all three host trains, whereas *Δ94* is unable to plaque on strains that do not supply exogenous copies of gp94.

### 2.4. Gene 94 Encodes Protein Containing a Putative N-Terminal Metal-Binding Motif and a C-Terminal HTH DNA-Binding Motif

Alexphander gene *94* is predicted to encode a 228 amino acid protein with a molecular mass of 26 kDa [10,26] (AlexPhander genome sequence GenBank Accession Number NC_051632.1). Both the MODELLER [27] and AlphaFold2 (AF2) [28] structure/function servers confidently predict that gp94 residues 143–226 comprise a three-helix winged helix–turn–helix DNA-binding motif resembling the MerR-type superfamily of HTH motif (Figure 4a,b). This motif is present in various DNA-binding proteins that perform numerous functions, including terminases (λ and P21 phages (Ortega et al. unpublished)), DNA-excision-related proteins [29], chromosomal anchor proteins (RacA [17]), bacterial regulators of phage packaging (Rpp [30]), and transcriptional regulators (CueR [18]). The DALI structural comparison server [31] indicates that the AF2-predicted N-terminal structure of gp94 has similarity to a DinB/YfiT superfamily protein from *Deinococcus radiodurans* that may contain a Zn^2+^-binding motif [32] (Figure 4c–f). The RMSD and DALI scores for the gp94 N-term alignment to the top structural ortholog are 3.22 and 7.7, respectively, suggesting a relatively high degree of confidence in the predicted fold (Figure 4c). In the N-terminal motif, α-helices 1 and 2 are connected by a loop with a low confidence structure, and the N- and C-terminal domains are linked by the α4-α5 loop (Figure 4a). The orientation between the N- and C-terminal motifs is not confidently predicted by AF2. Interestingly, the predicted MerR-type HTH is in the C-terminal region in gp94 but is N-terminal in closely related orthologs predicted by HHPred described above. The DRNApred server further predicts that gp94 residues N175, K176, R177, R178, Q180, T181, and R189 are involved in protein–DNA interactions—these residues align to the recognition helix in structural homologs [33]. Notably, a homology-based structural model of the gp94 C-terminal motif generated by MODELLER and AF2 aligns with the known wHTH motif-containing protein, λ Xis, with an RMSD of 1.26 Å (Figure 4b). In this model, gp94 residues N175, K176, R177, R178, Q180, T181, and K184 align to the major groove-binding surface and the positions of R177, Q180, T181, K184 align with positions of residues that are known to be important for Xis–DNA binding (E19, R22, R23, and R26, respectively; Figure 4b) [34]. These data are consistent with a role for the C-terminal motif of gp94 in DNA binding.

### 2.5. Gp94 Binds DNA Fragments and Phage Genomic DNA Nonspecifically Generating Cooperative Higher-Order Nucleoprotein Complexes

To investigate the biochemical properties of gp94, we purified the recombinant protein containing an N-terminal 6xHis-tag to >95% purity using Ni^2+^ affinity chromatography (Appendix A). Anti-His Western blot analysis confirms that the full-length His-tagged gp94 is the species that is purified via Ni^2+^ affinity chromatography and that gp94 migrates in SDS-PAGE at an apparent Mw of ~34 kDa, which is greater than its expected molecular weight of 26.0 kDa as calculated from its amino acid composition (Appendix A). Gp94 migrates as a 25–26 kDa monomer through a Superdex 200 gel-filtration column under native conditions (see methods), consistent with the molecular weight of a single protein chain of ~26.0 kDa with only a small peak corresponding to the predicted molecular weight of a dimer—about 54 kDa (Appendix A)—indicating that gp94 may function as a monomer or may form heterogeneous complexes with other proteins.

To evaluate the DNA binding properties of gp94, we performed native electrophoretic mobility shift assays (EMSA) using recombinant full-length gp94 and a C-terminal construct containing only the putative DNA binding motif (“C-term gp94”, residues 143–226). Each recombinant protein was engineered to contain an N-terminal 6xHis tag. A specific gp94 DNA-binding site has not been identified; therefore, we chose to measure binding to 500 bp fragments amplified from genomic loci adjacent to and within the gene *94* coding region extending 3 kb upstream (see Figure 2a for the genomic coordinates for fragments F1–F4). The rationale for choosing these sequences was that gp94 contains a MerR-superfamily HTH motif, and some proteins that contain this motif regulate their own transcription by binding to sites near their own coding locus [39,40,41]. It is noteworthy to point out that not all MerR-type HTH-containing proteins bind to sites near their coding locus, and a specific binding sequence, if there is one, may be located outside of the region tested.

EMSA data show that the full-length and C-terminal gp94 constructs both shift the mobility of fragments F1–F4 with similar profiles (Figure 5a,b representative of >3 independent trials). F1 and F2 DNA fragments are nearly completely shifted in the presence of 250 nM full-length and 162.5 nM C-term, and fragments F3 and F4 are completely shifted by 1000 nm full-length and 650 nM C-term (Figure 5a,b). Fragments F5–F10 cover 500 bp sites ranging 3 kb upstream of the gene *94* coding locus. These fragments showed the same binding profile as F3-F4. The gp94 binding profile was unaffected when varying the ionic strength or adding divalent metals such as Zn^2+^, Ca^2+^, Mg^2+^, and Cu^2+^.

We investigated whether the slight difference in binding affinity between F1 and F2 relative to all other fragments was due to a higher affinity site within these fragments. Here, we measured binding to F1 and F2 that are truncated by 100 bp sequentially from the 3′ end. These truncated fragments showed identical binding profiles relative to one another and the full-length fragments, indicating that there is no high-affinity site within the fragment and that gp94 binds dsDNA fragments with little sequence discrimination.

Neither full-length nor C-term gp94–DNA binding results in discrete, single protein–DNA complexes like those observed when fragment F1 DNA is incubated with the *Escherichia coli* nucleoid-associated protein, FIS (Figure 5c). Figure 5a,b indicate that, in some instances, there is some staining of DNA at the well interface, which itself disappears as more gp94 is added (compare lanes 3 and 4 of the F1-binding experiment in Figure 5b). This suggests that the gp94–DNA complex comprises multiple gp94 protomers. The apparent disappearance of the free DNA likely indicates that gp94 is forming multiple protein–DNA complexes wherein gp94 is coating or multimerizing on the DNA to generate a large molecular weight complex or that binding is sufficiently neutralizing DNA charge such that it does not enter the native enter the 6% 59:1 acrylamide:bisacrylamide gel. This binding profile resembles that of archaeal chromatin-associated proteins related to Sac7d [42] and a human viral protein inhibitor of cyclic GMP/AMP synthase [43]. The C-term DNA-binding motif alone also generates multiprotein–DNA complexes that are just barely able or unable to enter the gel, and the formation of high-molecular-weight complexes occurs with a high degree of cooperativity as few, if any, intermediate gp94–DNA complexes are observed, and free DNA is completely shifted over only a four-fold range of protein concentrations. Gp94 also forms large nucleoprotein complexes on purified phage genomic DNA (gDNA), whereas bovine serum albumin (BSA), a protein that is not known to bind DNA, did not shift genomic DNA in these assays (Figure 5d). Based on these observations, we hypothesize that gp94 binds to multiple sites within these 500 bp fragments with little sequence discrimination to “coat” the DNA molecule. It is also possible that gp94 can multimerize on DNA to generate higher-order nucleoprotein complexes, but we do not favor this possibility as the truncated C-term gp94 construct displays this cooperative formation of multiprotein complexes but lacks the proposed N-terminal multimerization motif.

### 2.6. Maximum Gene 94 Transcription Occurs Late during Lytic Growth

To investigate gene *94* transcription profiles, we performed mRNA expression analysis of gp94 and its neighboring genes to measure relative transcript quantities, identify the timing of transcription, and determine whether these genes are co-transcribed. Total mRNA was isolated from naïve *M. smegmatis,* Alexphander lysogen, and lytic phage infections at 0.5 and 2.5 h after adsorption. We then generated cDNA by reverse transcription and performed conventional PCR and qPCR analysis. Conventional PCR primers (Table 2) amplify products within genes *7* and *94* and across the boundaries of genes *91–96*, and qPCR primers amplify 60 pb segments within genes *7* and *94* (Figure 6a,b). We measured gene *7* expression as a control as it represents a gene known to be maximally expressed late in lytic growth from the late lytic promoter as observed by Ko et al., who used mRNA-seq to measure whole genome expression for the related F1 phage, Fruitloop [44]. PCR and qPCR amplification was dependent on reverse transcriptase and indicates that genes *7* and *94* are transcribed maximally late during lytic growth and that gene *94* is co-transcribed in an operon that includes genes *91–96*. This expression profile, while not exhaustive, does correlate well with that of other F1 phages for which mRNA-seq indicates that far-right-end genes are controlled by the late lytic promoter that also drives the expression of structural genes such as the major capsid protein [44]. We note that, in some cases, a very small amount of amplification is observed in the absence of reverse transcriptase, but that amplification is much more robust following reverse transcription. This could be due to low levels of Alexphander gDNA contamination, but it does not detract from the observation that amplification is clearly dependent on reverse transcription.

Genes 7 and 94 signals are present in samples prepared from lysogen cultures; however, we cannot distinguish whether this is due to expression from lysogen cells or the transition of some cells to lytic growth within the culture (induction). We further found that gene *94* expression increases during lytic growth relative to the levels observed in lysogenic cultures, peaking at 2.5 h post-infection. This pattern is similar to that observed for gene *7*, which encodes the major capsid protein, gp7 (Figure 6a,c). The expression of capsid from the lysogen culture may represent the tendency of the phage to induce within the culture growing at 37 °C or indicate leaky expression of lytic genes; again, our experiments cannot distinguish between these two possibilities. We do expect that lysogen cells can induce and release mature phage particles at some frequency under our growth conditions, given the ample zones of clearing around lysogen cells patched onto plates seeded with naïve *M. smegmatis* (Figure 1b). Gene *94* and *7* expression is maximal late in lytic growth 2.5 h after phage infection (Figure 6c). Overall, these findings, along with the essentiality of gene 94 for lytic growth, indicate a key role for gp94 during the lytic growth of phage Alexphander, which is likely mediated through its ability to bind dsDNA.

## 3. Discussion

Mycobacteriophage genomes are diverse and mosaic, and 40–60% of genes have unknown or uncharacterized functions. Here, we take a reverse genetics approach to elucidate the biological importance and biochemical features of phage protein gp94 in supporting the propagation of the temperate mycobacteriophage Alexphander. We report that gene *94* is essential for Alexphander lytic infections and is predicted to encode a protein that contains an N-terminal DinB-like motif and a C-terminal MerR-like superfamily HTH DNA-binding motif. We show that gp94 behaves as a monomer in solution and binds double-stranded DNA through its C-terminal DNA-binding motif to form large nucleoprotein complexes and that gene 94 transcription levels increase throughout lytic growth and peak at 2.5 h post-infection.

Alexphander is a cluster F1 phage that contains an integration cassette (genes *45*, *50*, and *55*) and forms lysogens on mc^2^ 155 at a frequency of 2.8%. The Alexphander lysogen displays robust phage release when patched onto naïve mc^2^ 155 lawns and is immune to superinfection by Alexphander and heterotypic infection by unrelated phages, LilPharaoh (cluster K1) and SchoolBus (cluster O). It is unlikely that heterotypic protection is mediated through the putative Alexphander immunity repressor, given how poorly related Alexphander is to LilPharaoh and SchoolBus at the nucleotide and gene content level. SchoolBus is a lytic phage that lacks an integration cassette, and the predicted LilPharaoh immunity repressor (gp38) is not related to the Alexphander immunity repressor (gp50) at the amino acid sequence level or in their predicted structures (ClustalW pairwise alignment score is 15.4). Thus, we hypothesize that the Alexphander prophage mediates defense against heterotypic attack by a yet uncharacterized, perhaps novel, system. A variety of such prophage-mediated defense mechanisms have been described for numerous host genera, including *Escherichia*, *Salmonella* [45], *Pseudomonas* [46], and *Mycobacterium* [47,48,49] phages. Future studies will investigate the immunity repression system in mycobacteriophage Alexphander and the possible role of prophage-expressed genes in mediating heterotypic protection.

Alexphander gene *94* (pham 165933) is encoded by numerous other actinobacteriophages; however, the percent identity and synteny of the gene *94* locus varies considerably between clusters (Figure 2b). For example, in F1 phages (e.g., Alexphander and Phasih), Alexphander gene *94* is located within a conserved four-gene locus that includes phams 815 (gene *93*/*89*), 2998 (gene *95*/*91*), and 371 (gene *96*/*92*) whereas these genes are absent in Clusters N and CZ phages, Schnauzer and BaxterFox, respectively (Figure 2b). Furthermore, in F1 phages, pham 165,933 encodes proteins that are 98 to 100% identical to one another but only 58–61% identical to cluster N variants at the amino acid sequence level. It is yet unclear if these differences indicate that AlexPhander gene 94 analogs have evolved subtle functional characteristics that are specific to each cluster. Functional variability may relate to DNA binding site preference, differences in protein–protein interacting partners, or differences in putative ligand specificities.

Alexphander gene *94* is annotated to encode a MerR-type HTH DNA-binding motif, a hallmark DNA-binding motif that is present in numerous DNA-binding proteins and was first discovered in a class of transcriptional regulators that modulates bacterial transcription in response to environmental stimuli such as metal ions and oxidative stress [40,50,51]. Notably, neither gene *94* nor other members of its gene phamily have been reported to play a role in transcriptional regulation through experimental validation, and given its unique domain architecture relative to MerR family transcriptional regulators and its essentiality for phage propagation under laboratory conditions, gp94 is likely to regulate a different DNA transaction that is essential for general propagation of the phage.

The AlphaFold 2 (AF2) structure prediction algorithm shows that the gp94 N-terminal domain adopts a four α-helix structure that is linked to the C-term wHTH domain by a long, possibly unstructured loop (Figure 4a). The gp94 N-terminal domain is structurally similar to the Dr0053 from *Deinococcus radiodurans,* a DinB/YfiT family protein, as determined from the DALI structural comparison algorithm [31]. Interestingly, the Dr0053 protein crystallizes with a robust dimerization interface between α1 and α4 of neighboring symmetry mates [32]. This suggests that gp94 may dimerize through its N-terminal domain; however, gp94 is predominantly in the monomeric state during size-exclusion chromatographic analysis (Appendix A). The Dr0053 crystal structure shows Zn^2+^ ions bound to the α2—α4 helices and at the dimer interface, which may indicate that gp94 acts as a monomer or that dimerization depends on metal ions, an unknown ligand, or forms heterodimeric complexes containing additional proteins. Zn^2+^ does not affect gp94–DNA binding, but we have yet to test whether the addition of metal affects the gp94 oligomeric state.

Gp94 can shift 500 bp DNA fragments; however, discrete complexes containing incremental increases in molecular weight are not observed as they are for the control protein, FIS (Figure 5a,c); instead, full-length and C-term gp94 variants bind 500 bp dsDNA fragments to generate multiprotein complexes that only barely enter a polyacrylamide gel (Figure 5a,b). The fact that both full-length and C-terminal gp94 constructs sequester free DNA to a higher-order nucleoprotein species suggests that this is mediated through a cooperative binding and coating mechanism driven by C-terminal DNA binding-motif interactions and not though multimerization events involving the N-terminal DinB/YfiT domain. Given the high degree of confidence with which gp94 residues 151–215 are predicted to adopt a wHTH motif, the DNA binding properties observed, and the fact that gp94 is essential for Alexphander lytic growth, we suggest that gp94 as an essential DNA-binding protein that is likely to regulate an essential DNA transaction through its ability to bind and coat DNA indiscriminately. This role could involve DNA packaging or phage chromosomal dynamics as displayed by terminases and chromosomal anchor proteins that contain a similar MerR superfamily HTH motif. We have yet to identify the gp94 consensus binding site, gp94 genomic targets, and/or gp94-associated proteins, all of which will inform a more detailed picture of the structural, energetic, and functional determinants that drive the essential role of gp94 in lytic phage propagation.

## 4. Materials and Methods

### 4.1. Strains and Plasmids

*Mycobacterium smegmatis* mc^2^ 155 and AP lysogens were grown in Middlebrook 7H9 medium supplemented with 10% Albumin Dextrose Complex (ADC; Dextrose, NaCl, Albumin Fraction V, Fisher Bioreagents, Whaltman, MA, USA) and 0.05% Tween 80, 50 µg/mL Carbenicillin, and 20 µg/mL cycloheximide as described previously [25], although tween was omitted and 1 mM CaCl2 included for phage infections. Plasmids used are listed in Table 1, and oligonucleotides in Table 1 and Table 2. Complementation strains and controls were grown in 7H9 lacking ADC but supplemented with 0.05% Tween 80, 0.2% succinate, and 20 μg/mL Kanamycin. Cultures were induced with 0.2% acetamide, as described below.

### 4.2. Lysogeny Efficiency

Lysogeny frequencies were measured by plating dilutions of *M. smegmatis* or AP lysogen (10^3^–10^7^ cfu) on agar plates seeded with 10^9^ pfu of phage Alexphander. Relative numbers of colonies on seeded vs. empty plates were determined, and the lysogeny efficiency was calculated as
(1)$$\frac{cfu\;per\;mL\;on\;phage\;seeded\;plates\;}{cfu\;per\;mL\;on\;clean\;7H10\;plates}\times 100$$

### 4.3. Constructing the 94 Deletions in Alexphander

Bacteriophage recombineering of electroporated DNA (BRED) was used to generate gene *94* deletions as described [24]. The gene 94 recombineering substrate was designed to eliminate nucleotides 52,527 to 53,159 (92% of the gene) but leaves the twenty-seven nucleotides at the 5′ and 3′ ends, which contain the stop codon of the overlapping gene *93* and potential regulatory sequences upstream of gene *95* (Figure 2c). The 400 bp deletion substrates were synthesized as gBlocks (IDT) and amplified using primers indicated in Table 2. The purified 400 bp deletion substrate was co-electroporated with Alexphander gDNA into an electrocompetent recombineering strain of *M. smegmatis* mc^2^ 155 using a gene pulser set to 2.5 kV, 1000 Ω, and 25 μF (Biorad, Hercules, CA, USA). The cell mixture was plated as a top agar lawn onto 7H10 plates with naïve *M. smegmatis* mc^2^ 155, mc^2^ 155 containing an empty expression vector (mc^2^ 155 (pLAM12), or mc^2^ 155 containing a complementation plasmid providing exogenous gp94 from an acetamide-inducible promoter (mc^2^ 155(pSH20)). The resulting plaques were screened for the presence of recombinant phages using flanking PCR. Flanking primers will generate 1240 and 640 amplicons for parental and recombinant deletion phages, respectively. Primary mixed plaques containing recombinant and wild-type phages were used to infect WT *M. smegmatis* mc^2^ 155 or a complementation strain containing gene 94 under the control of an acetamide-inducible promoter or an empty vector lacking the gene 94 coding region as described [24]. Hosts were grown in 7H10 lacking ADC and supplemented with 0.2% succinate, 0.05% Tween 80, 20 µg/mL Kanamycin to OD_600_ = 0.4–0.6. Cells were induced with 0.2% acetamide for three hours, and the induction media was exchanged with 7H9 supplemented with 0.2% acetamide, 1 mM CaCl_2_, and 20 µg/mL Kanamycin. To purify deletion phages, cells were incubated with primary plaque isolates for 30′ on ice and then plated as top agar lawns on 7H10 plates lacking ADC and supplemented with 0.2% acetamide and 20 µg/mL Kanamycin. Secondary plaques were picked as single plaques, in pools of 10, or as full plate lysates and analyzed by flanking PCR as described above.

### 4.4. Cloning, Expression, and Purification of Recombinant gp94

The full-length or C-terminal (residues 143–226) gp94 coding sequence, including a 6x-Histitdine tag added immediately after the gene 94 start codon (GTG), was cloned into the pET11a expression vector downstream of the T7 promoter element using the Gibson Assembly protocol [52] (New England Biolabs, Ipswich, MA, USA). Correct construction of plasmid pSH11 (full-length) and pSH19 (C-terminal) was verified by Sanger sequencing (Psomagen, Rockville, MD, USA). Plasmid constructs were transformed into the expression strain Nico21 (NEB). Transformants were streak purified and grown in 3 to 5 mL Luria broth (LB; Difco Detroit, MI, USA) and 50 µg/mL carbenicillin (Cb) at 37 °C overnight for 12 to 18 h (Acros organics Geel, Antwerp, Belgium). Either 200 mL or 1 L cultures of LB + Cb were inoculated 1:100 with starter culture and incubated at 37 °C until the OD_600_ reached 0.8 (~3–5 h). Cultures were kept at 37 °C or moved to a 30 °C incubator, and expression was induced with 0.2 mM Isopropyl β-D-1-thiogalactopyranoside (IPTG) for 4 h. Induction of full-length gp94 at 37 °C resulted in significant gp94 aggregation as determined by the amount of protein found in the cell pellet fractions thus subsequent induction was carried out at 30 °C. (Appendix A). Cells were harvested by centrifugation, resuspended in cell lysis buffer (50 mM sodium phosphate, pH 8.0, 0.30 M sodium chloride, 10 mM imidazole, 10% glycerol, and 1 mM phenylmethylsulfonyl fluoride (PMSF)), and lysed by sonication on ice for 200 s (20 s on and 20 s rest for 10 cycles). Cellular debris was cleared from the lysate by centrifugation at 10k rpm for 30 min, and genomic DNA was precipitated by adding polyethyleneimine (PEI) to a final concentration of 0.2%. The precipitated DNA was pelleted by centrifugation at 10K rpm for 30 min at 4 °C and the supernatant was applied to a column (Biorad) containing 3 mL bed volume of HIS-Select Ni^2+^ affinity gel resin (Millipore-sigma, Temecula, CA, USA) and allowed to flow through by gravity. The bound gp94-N-His constructs were washed with wash buffer (50 mM sodium phosphate, pH 8.0, 0.30 M sodium chloride, 20 mM imidazole, and 10% glycerol) and eluted with 2 mL each of elution buffer containing 50, 75, 100, 150, and 250 mM imidazole in 500 ul fractions. Fractions were analyzed for the presence of gp94 by SDS-PAGE, pooled, and dialyzed into storage buffer (50 mM sodium phosphate, pH 8.0, 0.30 M sodium chloride, 10% glycerol, 0.5 mM EDTA, and 1 mM DTT). Purified gp94 was quantified using the Bradford assay [53] and absorbance 280 on a nanodrop spectrophotometer (Thermo Fisher Scientific, Waltham, MA, USA), concentrated to 5 mg/mL using centrifugal protein filters (Millipore-Sigma), and analyzed for purity by SDS-PAGE.

### 4.5. Size-Exclusion Chromatography

Purified gp94 samples were subject to analytical size-exclusion chromatography (SEC) to ascertain the native oligomeric state. An amount of 500 µL of the sample at 5.0 mg/mL was loaded onto a Superdex 200 10/300 gl SEC column (Cytiva, Marlborough, MA, USA) that was pre-equilibrated with “storage buffer” (see above). A flow rate of 0.3 mL/min was maintained throughout the run, and protein retention was monitored by A280. The standard curve for protein molecular mass as a function of retention volume was generated using Gel Filtration Standard (Biorad) containing thyroglobulin (670 kDa), γ-globulin (158 kDa), ovalbumin (44 kDa), myoglobin (17 kDa), and vitamin B12 (1.3 kDa). An amount of 125 µL of 38 mg/mL Gel Filtration Standard was loaded onto the pre-equilibrated column, and standard retention volume was monitored as described above, plotted, and used to determine the molecular weight of gp94.

### 4.6. Bioinformatic Analyses

The gp94 amino acid sequence (GenBank Accession Number NC_051632.1) was obtained from phagesDB [10]. PSI-BLAST was used to identify proteins with similar sequences to gp94 from different databases [54]. Clustal omega was employed to perform multiple sequence alignments between full-length gp94 and C-term residues 151–215 and other wHTH-containing proteins [55]. The HHpred homology and structure prediction server was used to identify structural homologs of gp94 [56], and MODELLER and AlphaFold were used to generate template-based homology models of gp94 residues 151 to 215 [27,28]. TM-align was used to align the gp94 wHTH model to the Xis–DNA complex from an X-ray crystal structure of a FIS–Xis–DNA ternary complex (PDB ID: 6P0U). The DALI structural homology server was used to identify structural orthologs to the amino-terminal DinB/YfiT motif [31]. Molecular graphics images were generated with PyMOL [37].

### 4.7. DNA-Binding Assays

Electrophoretic mobility shift assays were used to measure gp94–DNA equilibrium binding to various DNA fragments or whole genomic DNA. Gp94-fragment binding was measured in binding buffer containing 0.1 to 0.5 M NaCl, 20 mM HEPES (pH 7.5), 5% glycerol, 0.5 mg/mL BSA, 1 mM EDTA, and 1 mM DTT (HEPES-BB). For the binding reaction, 50 ng of PCR amplified DNA fragment (~1.5 × 10^−7^ M) and 15.6 to 1000 nM (monomer) of purified full-length N-HIS-gp94 or 40 to 2600 nM (monomer) C-term gp94 were incubated on ice for 20 min and loaded on a pre-run, 6%, 59:1 acrylamide:bisacrylamide gel. Gels were run in 1X TBE running buffer (89 mM Tris (pH 7.6), 89 mM borate, and 2 mM EDTA) at 1 V/cm overnight at 4C or 10 V/cm at room temperature for 4 h. Gels were incubated with 1X SYBR gold or 1 μg/mL ethidium bromide and imaged with the azure 300 gel imager using the λ_ex_ = 472 and λ_em_ = 595.

Equilibrium binding to genomic phage DNA was performed in binding buffer containing 50 nM KOAc, 20 mM MgOAc, and 20 mM Tris-acetate, pH 7.9 (Acetate-BB). Genomic DNA (50–100 ng; 0.15 to 0.30 µM) was incubated with 100–1500 ng of full-length gp94 (192–2900 μM) or BSA and equilibrated at 37 °C for 20 min. Binding reactions were run on 7% agarose gels containing 1 µg/mL ethidium bromide, and gels were imaged with the Azure 300 gel imager using the λ_ex_ = 392 and λ_em_ = 595 filters.

### 4.8. RNA Analyses

Total RNA was isolated from wild-type *M. smegmatis* mc^2^ 155 cultures, AP lysogen cultures, and AP lytic infections both 0.5 h (early) and 2.5 h (late) post-infection. Three separate biological replicates were prepared for each growth phase. For lytic infections, wild-type *M. smegmatis* mc^2^ 155 cultures were grown to OD_600_ = 1.0 at 37 °C in phage infection media (see above). A total of 10 mL of culture was incubated with phages at an MOI = 10 for 30 min at room temp. Cultures were then incubated at 37 °C for the appropriate time prior to RNA isolation. For every 500 μL of culture, 1 mL of RNA protect reagent (QIAGEN, Hilden, Germany) was added. Samples were pelleted, and RNA was extracted using the RNeasy Mini Kit (QIAGEN). Cells were broken in tubes containing Matrix B beads (MP Biomedicals) using a “beadbeater” twice for 45 s on max speed with 1 min incubation on ice in between. Following DNaseI treatment (Invitrogen, Carlsbad, CA, USA), total RNA concentrations ranged from 0.5 to 1.0 µg/µL. cDNA was synthesized using random hexamers and maxima reverse transcriptase (Thermo Fisher); reverse transcriptase was omitted from “-RT” control reactions. Primers were designed to amplify within genes and across gene boundaries. Conventional PCR was performed with Phusion high-fidelity DNA polymerase (Thermo Fisher). qPCR primers were designed with the “Primer3” program [57,58], and qPCR was performed using the maxima SYBR green qPCR master mix and run on a CFX real-time PCR thermal cycler (Biorad) with thermal melt analysis enabled.

## Figures and Tables

**Figure 1 ijms-25-07466-f001:**
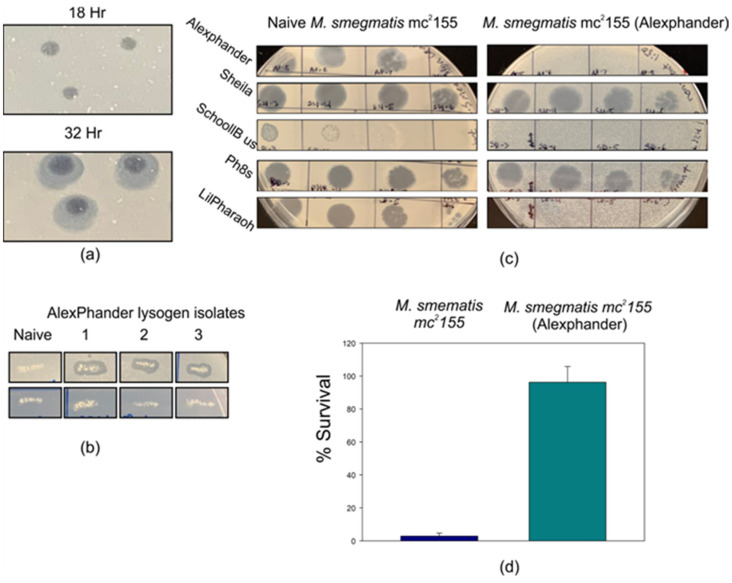
Mycobacteriophage Alexphander makes halo plaques and forms stable lysogens in Mycobacterium smegmatis mc^2^ 155. (**a**) Full plate infection assays show that mycobacteriophage Alexphander makes clear plaques on *M. smegmatis* that broaden and form halos after 32 h of growth. (**b**) Top row—phage release assay demonstrating phage release from Alexphander lysogen isolates (right three panels), but not from naïve *M. smegmatis* mc^2^ 155 strains (far left panel). These lysogen isolates are from the third and final round of lysogen purification for three independent tests. Bottom row—patched lysogen and naïve strain growth on canonical 7H10 solid agar plates. (**c**) Phage immunity assay indicates that the AP lysogen is immune to infection by phage Alexphander, phage SchoolBus (Cluster O), and phage LilPharaoh (cluster K1) but susceptible to infection by phages Sheila (cluster B1) and Ph8s (Cluster A2). The left panels show infection of the naïve host by five phages, from top to bottom: Alexphander, Sheila, SchoolBus, Ph8s, and LilPharaoh. The right panels show infection on AP lysogen by those same phages. Immunity and infection assays were repeated in triplicate, with each trial showing an identical infection profile. (**d**) Bar graph displaying the mean and standard deviation for three independent lysogeny efficiency tests for both naïve and AP lysogen strains. Lysogeny efficiency was calculated as $\frac{cfu\;per\;mL\;on\;phage\;seeded\;plates\;}{cfu\;per\;mL\;on\;clean\;7H10\;plates}\times 100$. AP lysogen strain is stable and able to grow on 7H10 agar plates that are seeded with Alexphander, while naïve *M. smegmatis* mc^2^ 155 is not.

**Figure 2 ijms-25-07466-f002:**
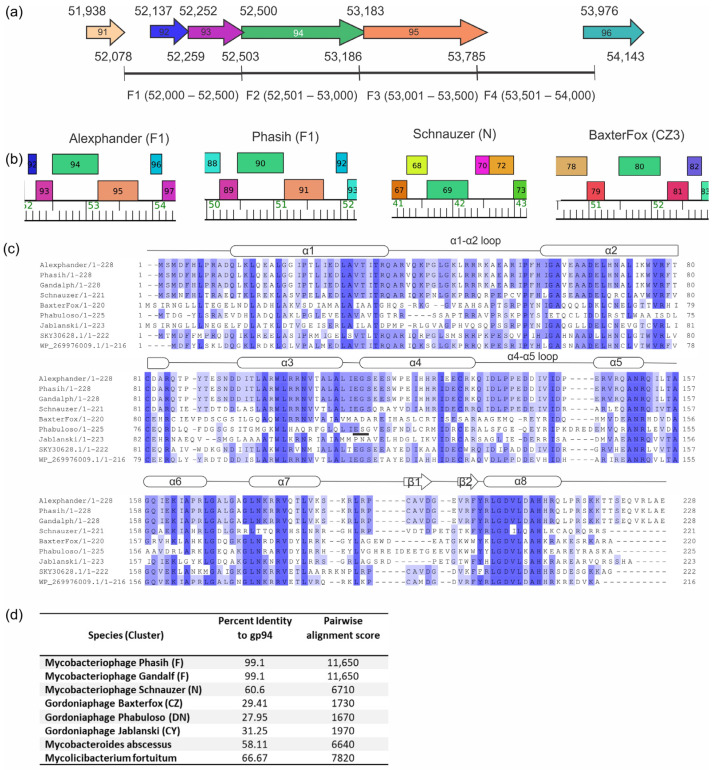
Gene *94* orthologs are present in mycobacteriophages and within other species of actinobacteria. (**a**) Gene *94* locus organization in Alexphander. Each gene is represented by a colored arrow that reflects the gene phamily to which the gene belongs; the direction of the arrow indicates the direction of transcription. Numbers above and below the gene represent the starting and ending coordinates of the reading frame, respectively. Four bases at the 5′ and 3′ end of gene *94* overlap with upstream *93* and downstream *95*, respectively. (**b**) The synteny of *94* is conserved within cluster F phages but diverges in cluster N mycobacteriophages and cluster CZ Gordonia phages. Green rectangles show the position of gene *94* in various genomes, whereas different colored rectangles represent the positions of neighboring genes. Gene numbers are shown. Genes with the same color rectangle belong to the same gene phamily. Phages Phasih, Schnauzer, and BaxterFox are generally representative of the gene synteny for phages in these clusters that contain a gene *94* ortholog. (**c**) Amino acid sequence alignment of Alexphander gp94 with orthologs. Amino acid conservation is indicated by the intensity of blue shading, with darker blue indicating amino acid conservation. The secondary structure of the predicted gp94 protein is shown as oblongs (α-helices) and arrows (β-strands). α6, α7, α10, β8 and β9 comprise the MerR-type HTH motif. (**d**) Pairwise alignment statistics between Alexphander gp94 and other orthologous proteins. Percent identity and the alignment score were calculated with Jalview using the BLOSUM62 scoring matrix [23].

**Figure 4 ijms-25-07466-f004:**
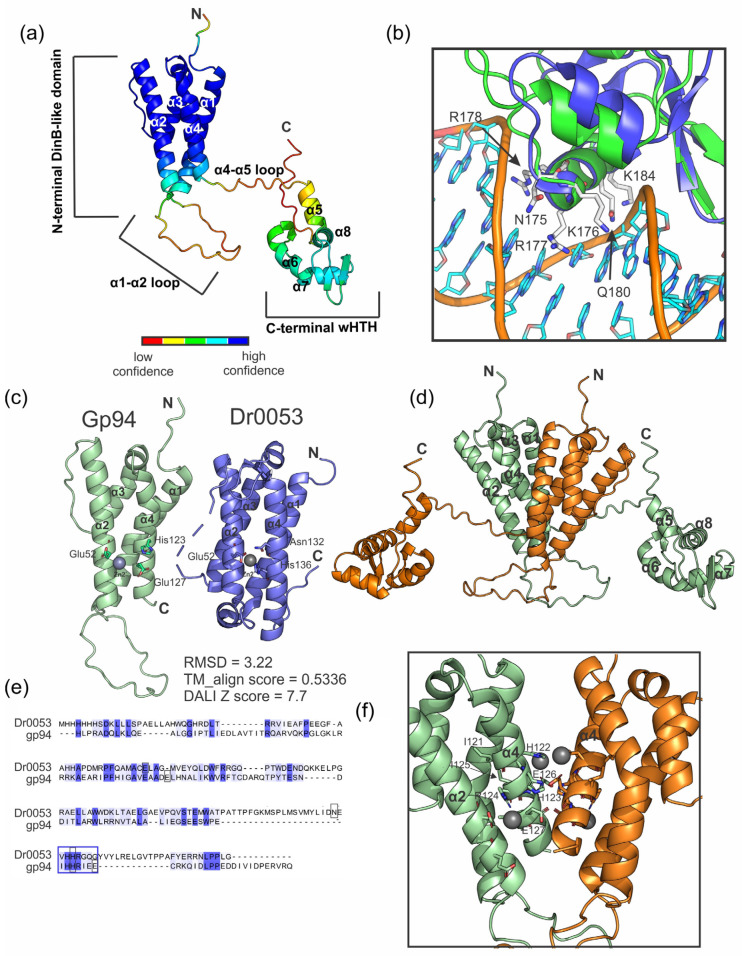
Alexphander gp94 is predicted to contain a C-terminal wHTH motif and an N-terminal DinB/YfiT domain. (**a**) AlphaFold2 prediction of the gp94 monomeric structure. The gp94 backbone is colored by confidence score as indicated, and relevant motifs are labeled. (**b**) Structural alignment of gp94 wHTH motif predicted by MODELLER (green) aligned with λ excise protein (blue) bound to DNA (orange backbone with cyan bases). The gp94 residues that align with the Xis recognition helix (N175, K176, R177, R178, Q180, and K184) are predicted to directly interact with the DNA major groove (PDBid 6P0U) [35]. (**c**) Structural comparison of the gp94 N-terminal motif (residues 1 to 131) and Dr0053 from *D. radiodurans* PDBID 6IZ2 [32]. The structures align with an RMSD and Tm-score of 3.22 Å and 0.577, respectively, according to the Tm-align structural alignment tool [36]. Zn^2+^ and proposed Zn^2+^-coordinating residues in the Dr0053 X-ray crustal structure and spatially analogous residues of gp94 are shown. It should be noted that Zn^2+^-binding properties have not been validated for Dr0053. (**d**) Structural model of AP gp94 dimer with each chain color-coded (green and orange). The dimer model was generated by aligning the N-terminal domain of individual gp94 monomers to the biological assembly of Dr0053 using the “super” algorithm in PyMOL [37]. (**e**) Pairwise alignment of Dr0053 and gp94 amino acid sequences calculated by Clustal W [38]. Darker blue shading indicates amino acid identities, and lighter blue represents amino acid conservation. Dr0053 residues that are potentially involved in Zn^2+^ binding and the spatially analogous gp94 residues are indicated with black boxes, and conserved residues located at the putative gp94 α4 dimer interface are indicated with blue boxes. (**f**) Structural model of AP gp94 dimer with each chain color-coded (green and orange) and potential dimer interface residues shown. Zn^2+^ atoms from the Dr0053 crystal structure that align to the putative gp94 dimer interface are also shown. Notably, Dr0053 has not been directly shown to bind Zn^2+^ [32], and gp94 lacks the canonical DinB/YfiT residues required for the canonical coordination of Zn^2+^.

**Figure 5 ijms-25-07466-f005:**
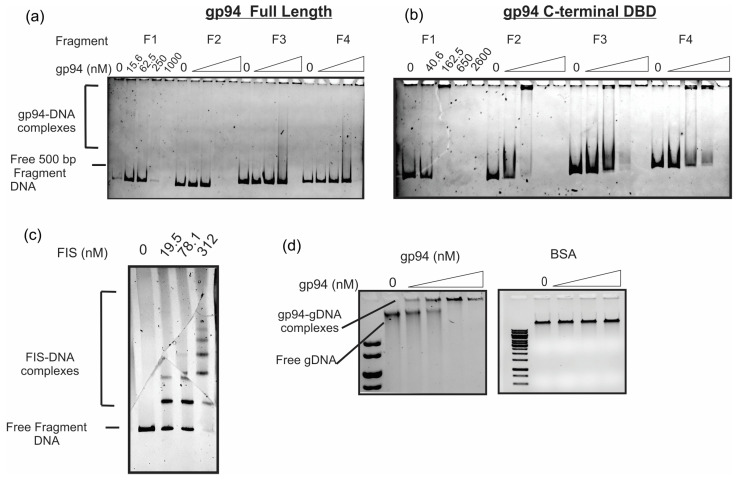
Gp94 binds 500 bp DNA fragments and genomic DNA. (**a**) Representative electrophoretic mobility shift assay (EMSA) showing gp94 binding to four different 500 bp DNA fragments corresponding to sequences neighboring or within the gene 94 locus (F1–F4). Free F1–F4 and protein-bound species are indicated. Genomic coordinates for fragment sequences are shown in Figure 2. For each fragment, 0, 15.5, 62.5, 250, or 1000 nM full-length gp94 was added to the DNA binding reaction, which was conducted in HEPES-BB + 0.1 M NaCl. (**b**) Representative EMSA as in “a” but with the C-term gp94 construct containing only C-terminal residues (143–226). Gp94-C-term monomer concentrations are given. Gels in (**a**,**b**) are representative of the binding profile observed for numerous experimental trials (>4). (**c**) EMSA for F1 fragment binding to the *E. coli* protein, Fis, a known DNA-binding protein that forms discrete protein–DNA complexes of increasing molecular weight by binding to related DNA sites with nM affinity. Fis was added to free DNA at 0 to 312 nM. (**d**) EMSA in 0.7% agarose gel with Alexphander genomic DNA in the absence and presence of increasing concentrations of gp94. Gp94 concentrations range from 0 to 1000 nM as in “b”. gDNA binding was measured in Acetate-BB and for gp94 (left panel) and BSA alone (right panel).

**Figure 6 ijms-25-07466-f006:**
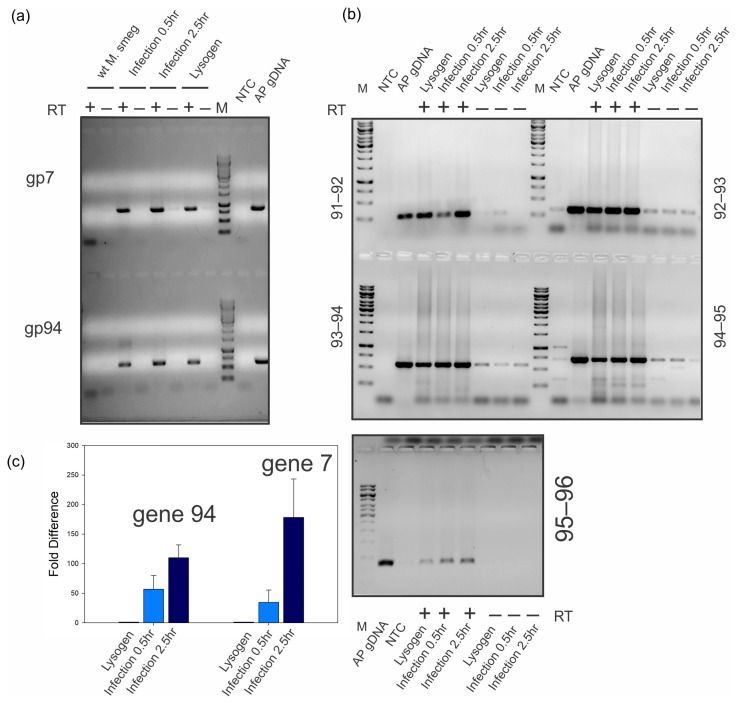
Gene 94 is maximally transcribed during lytic phage growth and is co-expressed with neighboring genes. (**a**) RT-PCR reactions were performed using primers that amplify within Alexphander gene *7* and gene *94* and were run on 0.8% agarose gels and stained with ethidium bromide. The presence of a gene-specific amplicon indicates the presence of gene-specific mRNA isolated from naïve *M. smegmatis* mc^2^ 155, early Alexphander lytic infections (0.5 h), late Alexphander lytic infections (2.5 h) and Alexphander lysogen cultures. The presence or absence of reverse transcriptase in the RT-PCR reactions is indicated. Lanes containing molecular weight markers, no template control, and positive control amplification from an AP gDNA template are indicated (M, NTC, and AP gDNA, respectively). (**b**) PCR of reverse-transcribed cDNA reactions was run using primers that amplify across neighboring gene boundaries. Cross-boundary amplicons suggest that a single contiguous mRNA containing both neighboring genes is present. Gene-boundary primers are indicated for each section of the relevant gel with marker and control lanes indicated as in “a”. Gels from (**a**,**b**) are representative of expression profiles resulting from 3 different biological replicates. (**c**) qRT-PCR for gene 7 and gene 94 expression. DnaA was used as the reference gene, and “fold-increase” represents gene-specific ΔCt values normalized to lysogen values. Fold difference values represent the mean for qPCR assays for three separate biological replicates. Error bars represent the standard deviation for all three replicates. No gene 7 or 94 signal was observed in naïve M. smegmatis mc^2^ 155 cultures.

**Table 1 ijms-25-07466-t001:** Plasmids used in this study.

Plasmid Name	Description	Source
pET11a	Parent expression strain	Novagen
pJV53	BRED recombineering plasmid	Van Kessel et al., 2007 [25]
pLAM12	Acetamide inducible complementation plasmid—parent	Marinelli et al., 2008 [24]
pSH11	pETt11a with N-HIS gp94 cloned downstream of the T7 promoter	This study
pSH20	Gene *94* complementation plasmid—acetamide inducible	This study constructed from pLAM12 [24]

**Table 2 ijms-25-07466-t002:** Oligonucleotides used in this study.

Purpose	Sequence	
Flanking PCR recombinant screening		
Forward primer	gttgtgcagtctgaacccaaccagg	
Reverse primer	gaccacaaacccgttcgctaagtg	
Gene *94* deletion substrate	tgccccattcaagaccctgacgagcaccgggcgttctgccaactccaagcagacatctacgcccacctcgccgacgtcccggcagaggtcggcgttgccgcagcggaactgctcgaacgccgcgagaaggaacgacgggaacagaaagcgatgttcaggaaggcattcgacaagtgagcatggacttccacctccccaggtctgaacaggtaagattagcagaatgagccgggtctttcgggtgcatctcaacgacgtcctcgctgcggaatgctgccaccccaactgctacgcgccagcccttactgacatagccagtcatgtgccgttgtgtgagcggcacatcatggttgtctaccgggaagccaatctcatgctcgccagccatagagctatggaa	
Deletion substrate amplification. Forward primer		
	tgccccattcaagaccc	
Reverse primer	ttccatagctctatggctgg	
Gibson assembly of Δ94 complementation construct		
Primer I1	gagatcggcggccgcatatgagcatggacttccacc	
Primer I2	gtcggaattcgccggggcgctcattctgctaatcttacct	
Primer O1	aggtaagattagcagaatgagcgccccggcgaattc	
Primer O2	gggaggtggaagtccatgctcatatgcggccgccgatctc	
Gibson assembly of full-length gp94 expression construct		
Primer I1	ttaagaaggagatatacatatgcatcatcaccatcaccacagcatggacttccacctcc	
Primer I2	ctgtccaccagtcatgctagctcattctgctaatcttacctgttcag	
Primer O1	ggtaagattagcagaatgagctagcatgactggtggac	
Primer O2	tggaagtccatgctgtggtgatggtgatgatgcatatgtatatctccttcttaaagttaa	
Gibson assembly of gp94 C-term expression construct		
Primer I1	ctttaagaaggagatatacatatgcatcatcaccatcaccacgacccggaacgggtc	
Primer I2	caccagtcatgctagctcattctgctaatcttacctgttc	
Primer O1	gacccgttccgggtcgtggtgatggtgatgatgcatatgtatatctccttcttaaag	
Primer O2	gaacaggtaagattagcagaatgagctagcatgactggtg	
PCR within and across Alexphander gene boundaries	Forward	Reverse
gene 7	gtgagcatggacttccac	tcattctgctaatcttacctgttc
gene 94	atggctgacatttcacgttcc	ttagctgccatccgggac
gb ^1^ 91–92	cacattgaccgggccaaag	ctggttgggttcagactgc
gb 92–93	caccgagcattccaccgac	cctgaacatcgctttctgttcc
gb 93–94	gaatgccccattcaaga	gcaagtcgatctgcttgcg
gb 94–95	gacgcaacgtgaccgcc	gaacgccaccatggatcgttc
gb 95–96	ctttcgggtgcatctcaacg	gtccaacagtttctccgcag
qPCR within gene boundaries		
gene 7	ggctgacatttcacgttccg	tcgaacacagtcgatccctt
gene 94	ctccgaagaatcctggcctg	tcttttctatctggccggcg

^1^ gb—indicated gene boundary.

## Data Availability

The authors confirm that the data supporting the findings of this study are available within the article. Any additional evidence of interest or raw image files may be requested from the corresponding author.

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
