# Peer review of "Mycobacteriophage Alexphander Gene 94 Encodes an Essential dsDNA-Binding Protein during Lytic Infection"

_ijms, 2024, doi:10.3390/ijms25137466_

Round 1

Reviewer 1 Report

Comments and Suggestions for Authors

The paper "Mycobacteriophages are viruses that specifically infect bacterial species within the genera Mycobacterium and Mycolicibacterium" provides a comprehensive investigation into the genomic diversity and functional analysis of the mycobacteriophage Alexphander, particularly focusing on the role of gene 94 in lytic infection. This study offers valuable insights into the molecular mechanisms of phage-host interactions. The experimental design is sound, and the results are compelling. The identification and characterization of gp94 as an essential DNA-binding protein, along with its role in forming multiprotein complexes on DNA, are significant contributions to the field.

Comments on the Quality of English Language

The quality of English and grammar is mostly fine. However, I detected a few issues and suggest the authors to perform a comprehensive reading of the manuscript to identify and correct any further issues.

Here are my suggestions:

Abstract, lines 22-26: The sentence is quite complex and dense. Breaking it down into two sentences might improve readability. Suggestion: "We hypothesize that gp94 is an essential DNA-binding protein for Alexphander lytic growth. It is proposed that gp94 forms multiprotein complexes on DNA through cooperative interactions involving its HTH DNA-binding motif at sites throughout the phage chromosome, facilitating essential DNA transactions required for lytic propagation."

Line 103: Remove the comma after Alexphander.

Line 105: Insert a space between 37 and °C.

Line 127: Remove the comma after 126.

Line 467: Remove the commas before and after gp94.

Author Response

Comments and suggestions for authors:

Comment 1: The paper "Mycobacteriophages are viruses that specifically infect bacterial species within the genera Mycobacterium and Mycolicibacterium" provides a comprehensive investigation into the genomic diversity and functional analysis of the mycobacteriophage Alexphander, particularly focusing on the role of gene 94 in lytic infection. This study offers valuable insights into the molecular mechanisms of phage-host interactions. The experimental design is sound, and the results are compelling. The identification and characterization of gp94 as an essential DNA-binding protein, along with its role in forming multiprotein complexes on DNA, are significant contributions to the field.

Response 1: The Authors thank you for your critical reading of the manuscript and for your helpful suggestions.

Comments on the Quality of English Language

Comment 1: The quality of English and grammar is mostly fine. However, I detected a few issues and suggest the authors to perform a comprehensive reading of the manuscript to identify and correct any further issues.

Here are my suggestions:

Response 1: Again, thank you for critical reading of the manuscript. We have taken your suggestions below and have re-evaluated the manuscript further for additional issues. 

Comment 1: Abstract, lines 22-26: The sentence is quite complex and dense. Breaking it down into two sentences might improve readability. Suggestion: "We hypothesize that gp94 is an essential DNA-binding protein for Alexphander lytic growth. It is proposed that gp94 forms multiprotein complexes on DNA through cooperative interactions involving its HTH DNA-binding motif at sites throughout the phage chromosome, facilitating essential DNA transactions required for lytic propagation."

Response 1: Thank you for this suggestion for clarifying the final sentence in the abstract. The authors have adopted the change that you suggest.

Comment 2: Line 103: Remove the comma after Alexphander.

Response 2: We have removed this comma.

Comment 3: Line 105: Insert a space between 37 and °C. 

Response 3: We have inserted this space.

Comment 4: Line 127: Remove the comma after 126.

Response 4: It has been removed.

Comment 5: Line 467: Remove the commas before and after gp94.

Response 5: We have removed these commas.

Reviewer 2 Report

Comments and Suggestions for Authors

Qui and colleagues tried to characterize the function of gp94 from a Mycobacteriophage, Alexphander. They found that gp94 governs phage propagation by regulating the genome transaction. The detailed molecular mechanism is not fully revealed in this work, but it provides novel insights into the specific Alexphander phage and paves the road for future studies on DNA-binding proteins from phages in general. I would recommend this manuscript to be published, if these questions/suggestions could be addressed:

Major issues:

In Figure 2a, gene 94 only overlaps 93. Also, Figure 2b would be more reader-friendly if the authors could enlarge the numbers of genes to the size of these shown in Figure 2A.

The legend of Figure 3b and c contains details about the method but lacks essential information about the corresponding figures. What gels are these? Are they just representative gels? In Figure 3c, it is reported that mc2155 (pSH20) had 9 positive results, but there are only three recombinant bands in Figure 3b. Also, Figure 3d is not discussed in the main text.

Section 2.5 could be deleted and Figure 5 could be moved to supplementary. The conclusion about a 28 kDa ‘minor’ variant of gp94 is not very convincing. In Figure 5c, the lanes 3 and 5 do not present their bands at the same intensity, meaning lane 5 could also have the 28 kDa band if more samples are loaded. Indeed, the chromatogram profile of Figure 5e shows that gp94 is perfectly monomeric and homogeneous, and the 56 kDa peak is negligible and could be random noise. Overall, in my opinion, the ‘minor’ band in the doublet is only a slightly degraded gp94 that always co-elutes with the major intact gp94. If the authors insist to keep this figure, please consider deleting Figure 5a, since 5b also contains these samples and the bands look clearer than these in 5a.

In Figure 6a, the supershift in the DNA-saturated full-length gp94 is not visible. It is clear that the free DNA bands are vanishing, but it would be really nice to have the shifted bands like what is shown in Figure 6b.

The authors’ interpretation about the gp94-DNA complexes being stuck at the wells (the last paragraph in page 13 and the final paragraph in discussion) is debatable. A gp94-DNA complex is very likely to be a nucleoprotein with no charge at the surface, making it immobile on a native acrylamide or agarose gel. Thus, the EMSA result does not indicate the large molecular weight of the complex.

Why did the authors choose gp7, a gene irrelevant to gp94, in section 2.7? Is it for supporting the conclusion about the late expression of gp94? It would be much appreciated if the rationale could be explained in this section. Also, could the authors elaborate on how citation 44 reports the late expression of a capsid protein?

Minor issues:

Line 169 only has six Gordoniaphage clusters.

Line 268 needs citation(s) for DNA binding terminase(s).

The a5 loop mentioned in line 277 is not present in Figure 4c.

Line 354, adjacent to

Line 390, Figure 6d

Line 434, similar to

Line 442, Figure 7c

Line 505, ‘motif’ is repeated; DNA binding proteins

Line 529, high molecular weight complexes

Comments on the Quality of English Language

‘However’ is not correctly used in lots of sentences. Please double check.

Author Response

Summary assessment: Qui and colleagues tried to characterize the function of gp94 from a Mycobacteriophage, Alexphander. They found that gp94 governs phage propagation by regulating the genome transaction. The detailed molecular mechanism is not fully revealed in this work, but it provides novel insights into the specific Alexphander phage and paves the road for future studies on DNA-binding proteins from phages in general. I would recommend this manuscript to be published, if these questions/suggestions could be addressed:

Response: Thank you for your critical review of the manuscript and your constructive feedback regarding revisions. The authors have made every effort to address each of the issues that you bring up in your review and we think that by addressing your comments, we have strengthened our manuscript. Thank you. Please see below for actions taken in addressing each comment.

Comment 1: In Figure 2a, gene 94 only overlaps 93. Also, Figure 2b would be more reader-friendly if the authors could enlarge the numbers of genes to the size of these shown in Figure 2A.

Response 1: Gene 94 does indeed overlap both genes 93 and 95. The ending coordinates of gene 94(53,186) and beginning coordinates of gene 95 (53,183) were swapped in the original figure. This has been corrected in the revised manuscript. Thank you for catching this. We have increased the font size of gene numbers in related phages.

Comment 2: The legend of Figure 3b and c contains details about the method but lacks essential information about the corresponding figures. What gels are these? Are they just representative gels? In Figure 3c, it is reported that mc2155 (pSH20) had 9 positive results, but there are only three recombinant bands in Figure 3b. Also, Figure 3d is not discussed in the main text.

Response 2: We have revised the legend to indicate that figure 3 b and d are representative agarose gels showing PCR reactions for various samples and that they indicate the presence and absence of the recombinant deletion allele which produced a shorter amplicon than the parental allele. Figure 3 b shows the presence of mixed populations when plated on a naïve host or a host containing a control empty plasmid (left and center panels) whereas the frequency of the and intensity of the recombinant allele in populations mixed populations increases when plated on the complementation strain. See the legend for more details.

Further, we note by asterisk in figure 3c that 9/35 plaques contained mixed populations when plated on the complementation strain and indicate that PCR for only three of these plaques are shown in figure 3b.

Finally, we have edited the main text to discuss figure 3d.

Comment 3: Section 2.5 could be deleted and Figure 5 could be moved to supplementary. The conclusion about a 28 kDa ‘minor’ variant of gp94 is not very convincing. In Figure 5c, the lanes 3 and 5 do not present their bands at the same intensity, meaning lane 5 could also have the 28 kDa band if more samples are loaded. Indeed, the chromatogram profile of Figure 5e shows that gp94 is perfectly monomeric and homogeneous, and the 56 kDa peak is negligible and could be random noise. Overall, in my opinion, the ‘minor’ band in the doublet is only a slightly degraded gp94 that always co-elutes with the major intact gp94. If the authors insist to keep this figure, please consider deleting Figure 5a, since 5b also contains these samples and the bands look clearer than these in 5a.

Response 3: Thank you for this suggestion. The authors believe that the temperature dependent solubility during expression, the aberrant SDS page migration, and the observed monomeric nature of the free protein are all notable properties of gp94 that we would like to report. We agree that the redox dependence of the minor “26 kDa” species in figure b is not very compelling and that this species could be a degradation product given the gels reported. We  have moved figure 5 to the supplementary materials and we have removed figure 5 panel b. Finally, we have altered the text to correspond to the figure renumbering and we have combined aspects of section 2.5 with the subsequent section 2.6 “Gp94 binds DNA fragments and phage genomic DNA nonspecifically generating higher-order nucleoprotein complexes.” We have kept Figure 5 panel a and b as they show the differential partitioning of the expressed gp94 protein under two different temperatures and therefore believe it is a notable observation.

Comment 4: In Figure 6a, the supershift in the DNA-saturated full-length gp94 is not visible. It is clear that the free DNA bands are vanishing, but it would be really nice to have the shifted bands like what is shown in Figure 6b.

Response 4: The authors agree that it would be desirable to see supershifted bands for full-length gp94 such as those that are observed for the C-term construct. These gels are representative and do not show bands corresponding to species entering the gel. This is typical of all gels we have run for the full-length construct under these conditions. This could be due the increased molecular weight of the complex owing to the N-terminal domain, or more substantial neutralization of the net charge of the bound DNA (although he pI of the N-terminal domain is 7.87). We have shown that the free DNA is not cleaved in kinetic assays that contain divalent metals such as Mg2+ and Mn2+ and see no degradation of the free DNA at a range of protein concentrations. We further ensure that the “disappearance” of DNA in our EMSA in figure 6 are likely not due to nuclease activity as our binding buffers contain EDTA and lack divalent metals that are required for the nuclease reaction mechanism.  

Comment 5: The authors’ interpretation about the gp94-DNA complexes being stuck at the wells (the last paragraph in page 13 and the final paragraph in discussion) is debatable. A gp94-DNA complex is very likely to be a nucleoprotein with no charge at the surface, making it immobile on a native acrylamide or agarose gel. Thus, the EMSA result does not indicate the large molecular weight of the complex.

Response 5: The authors agree that charge neutralization would influence the migration of DNA in the native acrylamide gel however numerous gp94 proteins would be required to significantly neutralize the DNA in our EMSA experiments resulting in the formation of a large molecular weight complex as stated by the authors. The DNA used in the EMSA experiments are 500pb with 1000 partially charged phosphates. The pI of the full-length protein is 9.54 with a high density of positively charged residues present at the proposed DNA binding interface. The predicted protein binding surface would directly interact with and neutralize the charge of ~ 9 phosphates and would contribute generally to a longer range neutralization of phosphate charge as determined by a poisson-boltzman potential distribution. Thus only ~1% of the charge is neutralized by a single binding event. To significantly neutralize charge, many gp94 protomers would need to bind. Thus the inability of gp94-DNA complexes to enter the native gel results from a combination of charge neutralization and molecular weight. The authors very much agree that the inability to enter the gel does not solely rest on the “size” of the complex but on its reduced negative electrostatic potential. This fact has been added to the interpretation of our EMSA data (lines 390-398) and at the end of the discussion (lines 546) where the term “large molecular weight” complex has been replaced with “multiprotein-DNA” complex, or higher-order nucleoprotein complex.   

Comment 6: Why did the authors choose gp7, a gene irrelevant to gp94, in section 2.7? Is it for supporting the conclusion about the late expression of gp94? It would be much appreciated if the rationale could be explained in this section. Also, could the authors elaborate on how citation 44 reports the late expression of a capsid protein?

Response 6: In section 2.7 We chose to look at the expression profile of gp7 because it is a known gene expressed during late lytic growth and is expressed under the control of the late lytic promoter so yes, it is to support the conclusion that gp94 and gp7 show similar timing of expression. We have included this in our discussion of these data (line 437).

Also, within this line we indicate that in reference 44, Ko et al use mRNA-seq to measure whole genome transcription from F1 phage, Fruitloop during lysogenic and lytic growth. These data show that the major capsid protein, along with other right arm genes, are transcribed maximally late during lytic growth.

Minor issues:

Line 169 only has six Gordoniaphage clusters.

Response: This is the correct number of Gordoniaphage that express a gene 94 ortholog. Thank you for catching this.

Line 268 needs citation(s) for DNA binding terminase(s).

Response: These are unpublished structures from the PDB. We have indicated this in the text.

The a5 loop mentioned in line 277 is not present in Figure 4c.

Response: Correct. readers should be directed to Figure 4a for an illustration of loop α4-α5. This has been clarified in the text.

Line 354, adjacent to

Response: This has been corrected.

Line 390, Figure 6d

Response: This has been corrected in the text, but this is now figure 5 and all references to this figure have been updated.

Line 434, similar to

Response: This has been corrected in the text.

Line 442, Figure 7c

Response: This has been corrected in the text, but this is now figure 6 and all references to this figure have been updated.

Line 505, ‘motif’ is repeated; DNA binding proteins.

Response: These have been corrected in the text.

Line 529, high molecular weight complexes

Response: These have been corrected in the text.

Comment: ‘However’ is not correctly used in lots of sentences. Please double check.

Response: We have checked this throughout the manuscript.